# Inclusion of Oat Polar Lipids in a Solid Breakfast Improves Glucose Tolerance, Triglyceridemia, and Gut Hormone Responses Postprandially and after a Standardized Second Meal: A Randomized Crossover Study in Healthy Subjects

**DOI:** 10.3390/nu15204389

**Published:** 2023-10-16

**Authors:** Mohammad Mukul Hossain, Juscelino Tovar, Lieselotte Cloetens, Anne Nilsson

**Affiliations:** 1Department of Food Technology, Engineering and Nutrition, Lund University, P.O. Box 124, 221 00 Lund, Sweden; juscelino.tovar@food.lth.se (J.T.); anne.nilsson@food.lth.se (A.N.); 2Division of Pure and Applied Biochemistry, Lund University, P.O. Box 124, 221 00 Lund, Sweden; lieselotte.cloetens@tbiokem.lth.se

**Keywords:** oats, polar lipids, glycaemic regulation, GLP-1, PYY, postprandial glycaemic response, bioactive components, RCT

## Abstract

Previously, it has been indicated that oat polar lipids included in a liquid meal may have the potential to beneficially modulate various cardiometabolic variables. The purpose of this study was to evaluate the effects of oat polar lipids in a solid food matrix on acute and second meal glucose tolerance, blood lipids, and concentrations of gut-derived hormones. The oat polar lipids were consumed at breakfast and effects on the biomarkers were investigated in the postprandial period and following a standardized lunch. Twenty young, healthy subjects consumed in total four different breakfast meals in a crossover study design. The breakfasts consisted of 1. White wheat bread (WWB) with an added 7.5 g of oat polar lipids (PLL); 2. WWB with an added 15 g of oat polar lipids (PLH); 3. WWB with and added 16.6 g of rapeseed oil (RSO) as a representative of commonly consumed oils; and 4. WWB consumed alone, included as a reference. All products with added lipids contained equivalent amounts of fat (16.6 g) and available carbohydrates (50 g). Rapeseed oil was added to the oat polar lipid meals to equal 16.6 g of total fat. The standardized lunch was composed of WWB and meatballs and was served 3.5 h after the breakfast. Test variables (blood glucose, serum insulin, triglyceride (TG), free fatty acids (FFA), ghrelin, GLP-1, PYY, and GIP) were measured at fasting and repeatedly during the 5.5 h after ingestion of the breakfast. After breakfast, PLH substantially lowered postprandial glucose and insulin responses (iAUC 0–120 min) compared with RSO and WWB (*p* < 0.05). Furthermore, a reduced glycaemic response to lunch (210–330 min) was observed following the PLH breakfast compared to all of the other breakfasts served (*p* < 0.05). Oat polar lipids (PLH) significantly reduced TG and ghrelin and increased circulating gut hormones GLP-1 and PYY compared to RSO (*p* < 0.05). The results show that exchanging part of the dietary lipids with oat polar lipids has the potential to improve postprandial blood glucose regulation and gut hormones and thus may have a preventive effect against type 2 diabetes.

## 1. Introduction

The global prevalence of obesity and cardiometabolic diseases (CMD) is increasing, and prevention strategies are urgently needed. The International Diabetes Federation (IDF) reported in 2022 that around 536 million people aged between 20–79 years were living with diabetes, and by 2045, the prevalence is predicted to rise to 780 million [1,2,3].

In this regard, the choice of diet is probably the most important lifestyle-related factor that can be used for the successful prevention of CMD. The botanical structure and composition of food, such as the contents of specific bioactive components, may influence the rate of digestion and absorption and may also influence post-absorption metabolism. Thus, it has been shown that the consumption of whole grains rich in several health promoting bioactive components is associated with a reduced risk of CMD and reduced body mass index (BMI) [4,5,6,7]. Oats are a source of various healthy bioactive components such as vitamins, minerals, antioxidants and phenolic compounds. It is well established that oats are efficient in lowering blood cholesterol concentrations. The cholesterol-lowering effect of oats is mainly attributed to their soluble dietary fiber (beta-glucans) [8,9]. Additionally, oats are high in polar lipids, which make up around 15 wt% of the total lipids [10,11]. The oat polar lipids include glycolipids, phospholipids and sphingolipids. The most abundant polar lipid in oats is the glycolipid digalactosyldiacylglycerol (DGDG) [10,11].

Studies investigating the effects of oat polar lipids on metabolic variables are scarce. However, recently it was shown that a liquid breakfast meal containing extracted oat polar lipids (12 g) significantly improved important CMD risk markers acutely after ingestion and, additionally, after a standardized meal consumed 3.5 h after breakfast [12]. Consequently, the observations revealed reduced postprandial blood glucose responses, increased concentrations of gut hormones involved in appetite and metabolic regulation, such as glucagon-like peptide 1 (GLP-1) and peptide YY (PYY), beneficial modulation of triglyceride (TG) concentrations, and decreased non-esterified fatty acids in young, healthy adults [12]. Furthermore, it has also been reported that the intake of liposomes made from oat oil resulted in an increased release of GLP-1 and PYY concentrations [13]. In parallel, subjective postprandial satiety sensation was sustained for 7 h, which contributed to reduced voluntary energy intake during the experimental day [13].

It is known that the food matrix (solid or liquid) may influence the rate of digestion and absorption of nutrients, and thus may also importantly affect the associated metabolic responses [14,15,16]. For instance, according to McClements et al., the presence of droplets in oil-in-water emulsions may have a marked influence on the digestion and absorption of lipids [17]. Chu and coworkers also reported that the presence of galactolipids in lipid droplets may inhibit lipolysis [18]. Studies on the metabolic effects of oat polar lipids have been performed in liquid or semi-liquid meal settings. In our previous study [12], high-pressure homogenization was applied during the test beverage preparation, which emulsified the oat polar lipids as microdroplets. To our knowledge, no research has been done yet on the influence of a solid food matrix on metabolic responses to oat lipids-containing meals.

The purpose of this study was to explore the metabolic effects of different doses (7.5 g and 15.0 g) of oat polar lipids incorporated in a solid breakfast meal. To achieve this objective, a randomized crossover meal study was conducted on a cohort of healthy young adults. Blood glucose, insulin, GLP-1, PYY, glucose dependent insulinotropic polypeptide (GIP), ghrelin and TG were measured in the postprandial period following the test breakfast and, additionally, after a subsequent standardized lunch. The effects of oat polar lipids were compared to equicaloric quantities of rapeseed oil. A breakfast meal without added lipids was included as a reference meal.

## 2. Materials and Methods

### 2.1. Study Participants

Twenty young, healthy subjects; 7 male and 13 female, with a mean age of 24 ± 0.24 years and a mean BMI 23 ± 0.49 kg/m^2^, participated in the meal study. The main inclusion criteria were an age between 20 and 40 years and a BMI between 19 and 28 kg/m^2^. Exclusion criteria were a fasting blood glucose concentration ≥ 6.1 mmol/L, any documented metabolic diseases or food allergies. Additionally, participants were required to be non-smokers. The administration of antibiotics or probiotics was prohibited for a period of two weeks before and during the trial period. Recruitments of test subjects took place between June and August 2020, and the clinical phase lasted from August to November 2020. Before being included in the study, every participant received a comprehensive explanation, both written and oral, of the objectives and methodology of the research. In addition, signed informed consent was collected from each participant. All participants were informed of their right to voluntarily withdraw from the trial at any point. The flow diagram of the study progress is available as Appendix A.

### 2.2. Test Breakfasts and the Standardized Lunch

A polar lipid-enriched oat oil (PL, 90% polar lipids, 10% TG) was specially prepared for the study and kindly provided by Swedish Oat Fiber AB (Bua, Sweden). This preparation was administered in two different doses as oil spreads prepared by mixing the selected oil volume with 10 mL water. A white wheat bread (WWB, Jättefranska, Pågen AB, Sweden) was included in the breakfasts both as a source of available carbohydrates to be consumed with the lipids investigated, and as a reference product without added lipids. The test meals (three in total) consisted of WWB with an added 1: 15.0 g PL (PL higher (H) amounts), 2: 7.5 g PL and rapeseed oil in ratio 50/50 (PL lower (L) amounts), or 3: 16.6 g rapeseed oil (RSO) poured directly on the bread. These three test meals contained equivalent amounts of fat (16.6 g). The RSO meal was used as a common oil reference. All breakfast meals contained 50 g available carbohydrates and were consumed together with 260 mL of water (Table 1)

The standardized lunch consisted of a meatball sandwich containing WWB, corresponding to 50 g available starch, and 100 g meatballs (Scan AB, Halmstad, Sweden). 250 mL of water was consumed in parallel. According to the nutritional information given by the manufacturer, the lunch meal contained a total caloric value of 485 kcal (Table 2).

### 2.3. Study Methodology and Design

The study was conducted using a single-blind, randomized crossover approach. Each test meal was consumed by all participants in a random sequence. Prior to each study visit (24 h), participants were instructed to refrain from engaging in vigorous physical activity, consuming alcoholic beverages, and consuming foods containing oats or high in dietary fiber (such as beans, whole grain bread, fiber-enriched pasta, and whole-cereal kernels). The participants were instructed to establish a standardized food routine before each study day. In order to ensure consistency, participants were instructed to provide a record of their dietary intake from the day preceding each study visit. In addition, the participants were given specific instructions to consume dinner meals that were standardized for each individual at 18:00 on the day preceding each study visit. Furthermore, they were instructed to consume a standardized evening meal at 21:00, which consisted of a commercially available white wheat bread with a topping of their choice. It is important to note that the same topping was used on all pre-experimental evenings.

The test products were provided in the form of breakfast meals. Each of the products was tested with a one-week interval between each test. The participant arrived at the study center at 07:30 after 10 h of fasting during the night. Capillary blood samples were collected, followed by the consumption of a test meal at time zero (0 min), with a designated consumption time of 10–12 min. Subsequent capillary blood samples were collected at specific time intervals of 15, 30, 45, 60, 90, 120, 150, 180, and 210 min after the start of the breakfast meal. Following the blood test conducted at the 210-min mark, a standardized lunch was provided, and further blood samples were collected at intervals of 225, 240, 255, 270, 300, and 330 min following the start of breakfast. The amount of blood drawn at a specific sampling point did not exceed 800 µL (for insulin, 200 µL; for TG and gut hormones combined, 600 µL). That equals a total of 5 mL of blood per subject per study day. Throughout the duration of the trial, the participants were confined to the clinical facility and were strictly prohibited from consuming any food or beverages, with the exception of the breakfast and lunch meals that were provided to them. The individuals were instructed to minimize their engagement in physical activities to the greatest extent feasible.

### 2.4. Biomarker Assessment

The measurement of all biomarkers was carried out using capillary blood samples. Plasma glucose concentrations were measured in whole blood at the time intervals specified above, using a HemoCue Glucose 201+ analyzer manufactured by HemoCue AB in Ängelholm, Sweden. Samples for the analysis of serum insulin, free fatty acids (FFA), and TG were obtained using BD Microtainer SST tubes. The serum insulin samples were taken at the same time points as the glucose determinations, except for 15 and 150 min. The concentrations of FFA and TG were measured at 0, 60, 120, 210, 270, and 330 min. The tubes were allowed to settle at room temperature for about 30 min prior to centrifugation for a duration of 5 min. The centrifugation was performed at a speed of 5000 revolutions per minute (rpm) at a temperature of 25 °C, using an Eppendorf centrifuge model 5425. Subsequently, the serum was separated and kept at a temperature of −40 °C until it was subjected to analysis.

Blood plasma samples were taken for the analysis of total concentrations of GLP-1, GIP, PYY, and ghrelin at four different time points (0, 60, 210, and 330 min). The samples were collected using BD Microtainer K2E tubes. A mixture of a DPP-4 inhibitor (10 µL/mL blood, Millipore, St. Charles, IL, USA) and aprotinin (50 µL/mL blood, Sigma-Aldrich, St. Louis, MO, USA) was injected into the tubes as an inhibitory cocktail before the blood collections. The tubes were stored on ice prior to and during the collection of samples. Subsequently, the tubes were subjected to centrifugation for a duration of 10 min at a speed of 4200 rpm and a temperature of 4 °C. This centrifugation process was carried out within a time frame of 30 min after the blood was collected. The plasma was thereafter separated, and samples were subjected to freezing at a temperature of −40 °C until the time of analysis.

The measurement of insulin concentrations was conducted using a solid phase two-site enzyme immunoassay kit (Insulin ELISA 10-1113-01, Mercordia AB, Uppsala, Sweden). The amounts of FFA were determined using an enzymatic colorimetric approach on a 96 microplate. Specifically, the NEFA-HR (2) ACS-ACOD method provided by FUJIFILM Wako Chemicals Europe GmbH in Germany was used for this analysis. The amounts of serum TG were measured using a multi-sample enzymatic assay known as LabAssay™ Triglyceride 290-63701, which follows the GPO.DAOS technique. This test was conducted by FUJIFILM Wako Chemicals Europe GmbH, Germany. The measurement of total plasma concentrations of GLP-1, PYY, GIP, and ghrelin was conducted using a 10-spot U-plex test kit (Meso Scale Diagnostics LLC, Rockville, MD, USA). In accordance with the suppliers’ description, biotinylated capture antibodies are coupled to U-PLEX Linkers. The U-PLEX Linkers then self-assemble onto unique spots on the U-PLEX plate. After analytes in the sample bind to the capture reagents, detection antibodies conjugated with electrochemiluminescent labels (MSD GOLD SULFO-TAG) bind to the analytes to complete the sandwich immunoassay.

### 2.5. Calculation and Data Analysis

The data are presented as means ± SEM. The incremental areas and total areas under the curves (iAUC and AUC, respectively) were calculated using a trapezoidal model for every participant and test meal. The iAUC was calculated for statistical analyses of blood glucose and insulin concentrations. AUC was used for assessing the concentrations of TG, GLP-1, PYY, GIP, and ghrelin. The graph plotting and area calculation were conducted using GraphPad Prism (version 9.4, GraphPad Software, La Jolla, CA, USA).

The randomization of the consumption sequence of the test meals was achieved using the randomization features available in Microsoft Excel (Washington, DC, USA). The variations in the results among different products (‘Meal’: PLH, PLL, RSO, and WWB) at different times throughout the experimental day (‘Time’) were assessed using a mixed model approach (PROC MIXED in SAS release 9.4; SAS Institute Inc, Cary, NC, USA) with repeated measures and an autoregressive covariance structure for the test variables. The subjects were treated as a random variable, with the associated baseline (fasting values) included as a covariate in the model. The physiological responses resulting from the test products were assessed using ANOVA (general linear model) followed by Tukey’s pairwise multiple comparison test in MINITAB Statistical Software (version 18.1, Minitab, Minitab Inc, State College, PA, USA). The significance level was set at a *p*-value threshold of less than 0.05.

### 2.6. Power Calculation

The primary outcome measure of the study was incremental blood glucose concentration changes, iAUC, 0–120 min after the breakfast meal. Assuming a difference of 22 mmol·min/L (15%) between test meals and a SD of 72 mmol·min/L, with α = 0.05 and 1 − β = 0.8 [12], 15 test subjects were required. We decided to increase the number of test subjects to 20 to cover eventual dropouts according to common rates.

## 3. Results

### 3.1. Baseline (Fasting) Characteristics

No statistically significant differences were seen in concentrations of any of the determined test variables at fasting (i.e., before the start of test meals; *p* > 0.05). Data can be found in Table 3 (glucose and insulin), Table 4 (TG and FFA), and Table 5 (Ghrelin, GLP-1, PYY, and GIP).

### 3.2. Glucose and Insulin

The study results indicated that meals exhibited a significant main effect (0–330 min) on glucose responses as well as meal·time interactions (Figure 1). The iAUC (0–120 min) of the glucose curve following the breakfast meal with PLH was significantly reduced compared to WWB (*p* < 0.001) and RSO (*p* < 0.01). Furthermore, the breakfast postprandial glucose concentrations (iAUC 0–120 min) were reduced following the breakfast including PLL compared to the breakfast containing WWB (*p* < 0.01). After the standardized lunch (iAUC 210–330 min), glucose responses were significantly decreased after the PLH breakfast compared to after the breakfasts composed of WWB (*p* < 0.001), PLL (*p* < 0.01), and RSO (*p* < 0.01), Table 3.

A main effect of each meal was observed on insulin responses during the experimental period (Figure 2). The results revealed significantly lower postprandial insulin responses (iAUC 0–120 min) after breakfast following PLH compared to the breakfast with WWB (*p* < 0.001) and RSO (*p* < 0.01), and the insulin responses to the standardized lunch (iAUC, 210–330 min) were significantly lower after consuming the PLH breakfast compared to the RSO (*p* < 0.05, Table 3).

### 3.3. Triglycerides and FFA

The significant main effect of meals on TG concentration was found over the test period (0–330 min, Figure 3). The TG responses after breakfast (AUC 0–210 min) and lunch (210–330 min) were significantly lower after intake of PLH compared to after RSO (*p* < 0.05, Table 4). As expected, the WWB breakfast resulted in the lowest concentration of TG during the test period, however, no significant differences were detected in TG concentrations after intake of WWB compared with PLH.

There were no significant variations found in FFA concentrations during the whole experimental period (*p* > 0.05, Figure 4, Table 4).

### 3.4. Ghrelin

The results showed the significant main effect of products over the test period on ghrelin concentrations (Figure 5). Ghrelin concentrations post breakfast (AUC 0–210 min) were significantly lower after the PLH and PLL breakfasts compared to the RSO and WWB breakfasts (*p* < 0.05).

### 3.5. GLP-1

The results are presented in Figure 6 and Table 5. A significant main effect of products was detected along the test period (0–330 min, *p* < 0.05). The GLP-1 concentrations (AUC) in the time period 0–210 min were significantly increased after the breakfasts containing PLH and PLL compared to the WWB and RSO breakfasts (*p* < 0.05). The GLP-1 concentration at 330 min was significantly higher after the PLH breakfast compared to after the RSO and WWB breakfasts (*p* < 0.05).

### 3.6. PYY

Results concerning PYY concentrations are shown in Figure 7 and summarized in Table 5. A significant main effect of the test products on PYY concentrations was observed along the test period (0–330 min, *p* < 0.05). The PYY concentrations after the PLH breakfast (AUC 0–210 min) exhibited a statistically significant increase in comparison to both the WWB (*p* < 0.001) and RSO breakfasts (*p* < 0.05). The increase in PYY concentrations after consuming a PLH breakfast persisted after consuming a standardized lunch. As a consequence, the PYY concentrations were considerably higher after the PLH breakfast meal compared to the WWB and RSO breakfasts at the end of the test period (at 330 min, *p* < 0.01).

### 3.7. GIP

The postprandial outcomes of GIP after the consumption of breakfast and standardized lunch meals are shown in Figure 8 and summarized in Table 5. There were no significant main effects of the products on GIP concentrations over the study period (0–330 min). However, the results indicated a significant meal·time interaction, exhibiting that the breakfast consisting of PLH led to decreased postprandial GIP concentrations over the 0–210 min compared to the RSO and PLL meals (AUC, *p* < 0.001).

## 4. Discussion and Conclusions

The present work studied the effects of two different doses of oat polar lipids (15.0 g and 7.5 g) incorporated into a solid food matrix (bread) on cardiometabolic test variables. The biomarkers were investigated in the postprandial period after the breakfasts and also after a subsequent standardized lunch meal. Our results revealed that 15.0 g, and to a minor extent also 7.5 g, of oat polar lipids may improve acute and second meal glucose tolerance, increase gut hormones important in appetite and glycaemic regulation, and improve the postprandial blood lipid profile compared to both an isolipidic amount of a rapeseed oil-containing meal and/or a close to fat-free reference meal.

The results from the present study support previous observations from this research group demonstrating the health effects of 12 g of oat polar lipids as part of a carbohydrate-rich liquid breakfast meal [12]. However, in the present study, a highly concentrated polar lipid preparation (90% polar lipids) was used to obtain a significant decrease in the total fat content of the test meals compared to the previous study. Thus, the total fat content in the test meals investigated in the current study was 16.6 g, compared with 33 g in the previous one [12]. The lower total fat content in the meals, as well as differences in the food matrix (solid vs. liquid) are factors that may modify the impact of a bioactive on cardiometabolic biomarkers. The decreased total fat content in the present study increases the possibility of drawing conclusions about the specific effects of polar lipids per se on cardiometabolic risk markers, minimizing the general effect of the intake of fat on these indicators.

In the present investigation, we found that the high dose (15.0 g) of oat polar lipids significantly improved most of the variables assessed in comparison to rapeseed oil, whereas only a few variables managed to reach a statistically significant improvement after the low dose of oat polar lipids (7.5 g). However, it is noteworthy that, in comparison to rapeseed oil, the 7.5 g dose significantly increased the gut hormone GLP-1 and significantly decreased the hunger sensation hormone ghrelin. When compared to WWB, the low dose of polar lipids significantly decreased glycemia; nevertheless, it was not significantly different when compared to rapeseed oil. It would be interesting to investigate the effects of PL administered at doses lying between those used in the current study.

Both PLH and PLL resulted in significantly decreased acute postprandial glucose responses after the breakfast compared to WWB, whereas no significant effects were observed after RSO. The PLH meal also resulted in a significantly reduced glucose response compared to the RSO meal. These results show that not all lipids are equally potent to lower the acute postprandial glucose concentrations. In addition, the results demonstrate that a dose equal to 7.5 g of oat polar lipids is sufficient in this respect. Interestingly, the PLH breakfast also resulted in improved glucose tolerance after the standardized lunch meal. The insulin responses followed the same pattern as glucose. Consequently, the PLH breakfast resulted in lower postprandial insulin concentrations after both the breakfast and the standardized lunch periods compared to the WWB and RSO breakfasts. The beneficial modulation by oat polar lipids of second meal glucose tolerance is an interesting health effect. These observations confirm the glycemia-modulating properties of oat polar lipids that goes beyond the known action of dietary fats in general [19].

Food matrices play an important role in the bioavailability and absorption of nutrients. Numerous studies have shown that the food matrices, calorie content, kinds of accessible macronutrients, volume, and complexity of the meal all have an impact on the degree to which the macronutrients are digested [16,20,21,22,23]. Solid meals require more mechanical digestion, which slows down the process of nutrient absorption and causes, for instance, a slower and more prolonged increase in blood glucose levels. According to several reports, the rate of stomach emptying is noticeably slower for solid meals than for liquid meals [24,25], which can cause the macronutrients to be digested and absorbed more slowly. The rate of the starch digestion affects the postprandial rise of blood glucose and, consequently, insulin responses. However, in the present study, the solid food matrix did not alter the overall output of the study results in respect to the glycaemic response compared with a previous study that used a liquid food matrix [12]. The overall number of calories from dietary fat in the current trial was approximately half (149 kcal vs. 297 kcal) that in the previous study, which may have reduced the generic influence of fat on the digestion and absorption rates of macronutrients. Furthermore, all fat-enriched meals had an equal amount of dietary lipids, which strengthens the suggestion that oat polar lipids were the components responsible for the postprandial glycaemic response decrease.

In accordance with previous observations [12], the oat polar lipids increased the release of GLP-1 and PYY. Thus, the results regarding the effects of oat polar lipids on the release of these gastrointestinal hormones can be regarded as robust. This is an important finding, since gastrointestinal hormones have beneficial implications in glycaemic and appetite regulation. Consequently, the increase of GLP-1 and PYY was observed in parallel to the improved postprandial glucose regulation.

The mechanisms behind the beneficial modulation of glucose tolerance by oat polar lipids are not yet clear. Ohlsson et al., in their study on oat polar lipids-enriched liposomes, suggested a slow digestion of these lipids in vivo, which could potentially result in an enhanced release of gut hormones in the gastrointestinal tract [13]. In the human intestine, the digestion of galactolipids is carried out by pancreatic lipase-related protein 2 (PLRP2) [26]. However, the delayed hydrolysis of polar lipids by pancreatin has been observed in vitro, suggesting a similar effect in vivo [27]. This may lead to delayed absorption, which could potentially result in an enhanced release of GLP-1 and PYY. These gut hormones are potentially involved in various metabolic processes, such as the modulation of gastrointestinal motility and gastric emptying rate.

GIP appears to have a bifunctional role, promoting both insulin secretion and adipogenesis, which potentially contribute to weight gain and the development of metabolic disorders [28,29]. Remarkably, our findings revealed that the PLH breakfast resulted in lower postprandial GIP concentrations compared to the RSO and PLL breakfasts in the breakfast postprandial period. The consequences of reducing GIP release are difficult to interpret and require further investigation.

In the present study, ghrelin was significantly reduced in the postprandial period (0–210 min) after both the PLH and PLL breakfast meals compared to the WWB and RSO. Ghrelin stimulates appetite and accelerates gastric emptying, which triggers hunger sensation [30]. This indicates that oat polar lipids in solid meals may potentially reduce the hunger sensation, and the reduced ghrelin concentration before commencing the lunch (at 210 min) may potentially lead to reduced calorie intake during the meal.

As energy providers and carriers of dietary fat, TGs are essential to metabolism. Postprandial increased TG have, however, been suggested among cardiometabolic risk factors [31]. In contrast to meals containing rapeseed oil (RSO), our findings demonstrated a considerable decrease in postprandial triglyceridemia after the consumption of a breakfast with oat polar lipids (PLH). Thus, the possibility that oat polar lipids can lower postprandial triglyceridemia and hence lower the risk of cardiovascular illnesses is strengthened by this research.

To fully understand the underlying mechanisms behind these encouraging health benefits of oat polar lipids, further research is required. Since the PLL breakfast resulted in some, but limited, beneficial effects compared with the PLH, it would be interesting to investigate the dose-response effects of PL with doses in between the PLL and PLH breakfasts to find a dose as low in total fat as possible, but still efficient in improving cardiometabolic risk variables. Although the results indicate the metabolic beneficial effects of a low dose (7.0 g) of oat polar lipids on postprandial insulin and ghrelin responses, a dose closer to 15.0 g seems necessary to obtain more convincing effects on postprandial glycaemia, TG and other gut hormone responses. Future studies should also investigate whether the present observations with oat polar lipids can be extrapolated to plant polar lipids in general. It would be interesting as well to explore the long-term effects of oat polar lipids.

The current study had two major limitations. The first one relates to the collection of capillary blood, which limited the sample volume. This, in turn, restricted the possible number of test points and test markers to be evaluated. Moreover, the sample size of the study was calculated to detect significant changes in blood glucose levels. However, the sample size might lack statistical power for the secondary outcomes. Studies in other populations, larger cohorts, and longer-term trials are desirable to confirm the results.

In conclusion, this research demonstrates the beneficial effects of polar lipids from oats on acute and second meal postprandial glycaemic control, blood lipids, and gastrointestinal hormones in healthy volunteers. The results suggest that these lipids may have anti-obesogenic and anti-diabetic properties, and thus may be included in innovative foods with the purpose of preventing cardiometabolic illnesses. The recorded effects are probably independent of the type of food matrix (solid or liquid).

## Figures and Tables

**Figure 1 nutrients-15-04389-f001:**
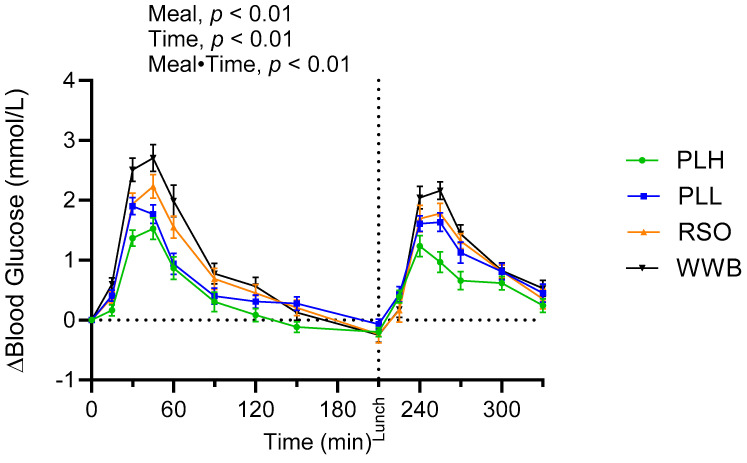
The incremental changes in blood glucose levels after the consumption of test breakfasts and standardized lunch meals. The values are expressed as means ± SEM, with a sample size of n = 20. Repeated measures; mixed model in SAS. WWB, white wheat bread; PLH, 15.0 g oat polar lipids; PLL, 7.5 g oat polar lipids and 8.3 g rapeseed oil; and RSO, 16.6 g rapeseed oil.

**Figure 2 nutrients-15-04389-f002:**
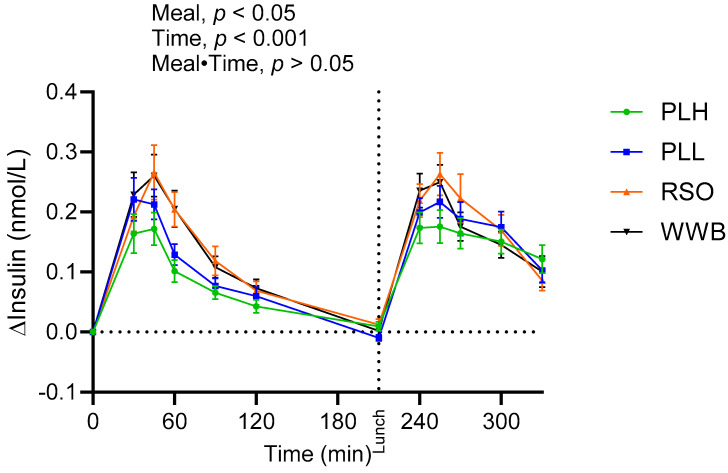
Incremental changes in serum insulin concentrations after test breakfasts and standardized lunch meals. Values are expressed as means ± SEM, with a sample size, of n = 20. Repeated measures; mixed model in SAS. WWB, white wheat bread; PLH, 15.0 g oat polar lipids; PLL, 7.5 g oat polar lipids and 8.3 g rapeseed oil; and RSO, 16.6 g rapeseed oil.

**Figure 3 nutrients-15-04389-f003:**
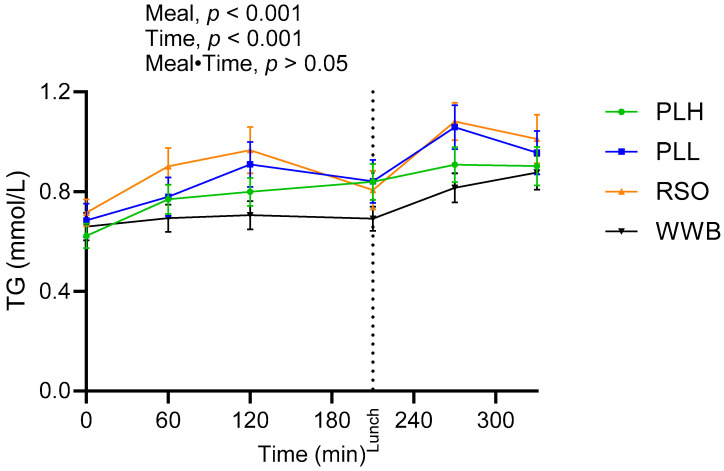
Concentration of serum triglycerides (TG) after breakfast with the oat polar lipid enriched test products and the standardized lunch meals. Values are expressed as means ± SEM, with a sample size of n = 20. Repeated measures; mixed model in SAS. WWB, white wheat bread; PLH, 15.0 g oat polar lipids; PLL, 7.5 g oat polar lipids and 8.3 g rapeseed oil; and RSO, 16.6 g rapeseed oil.

**Figure 4 nutrients-15-04389-f004:**
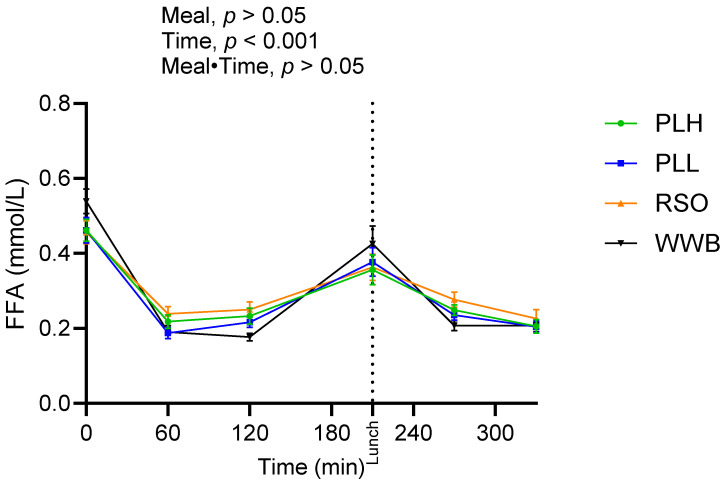
Concentration of serum FFA after breakfast with the oat polar lipid-enriched test products and the standardized lunch meals. Values are expressed as means ± SEM, with a sample size, of n = 20. Repeated measures; mixed model in SAS. WWB, white wheat bread; PLH, 15.0 g oat polar lipids; PLL, 7.5 g oat polar lipids and 8.3 g rapeseed oil; and RSO, 16.6 g rapeseed oil.

**Figure 5 nutrients-15-04389-f005:**
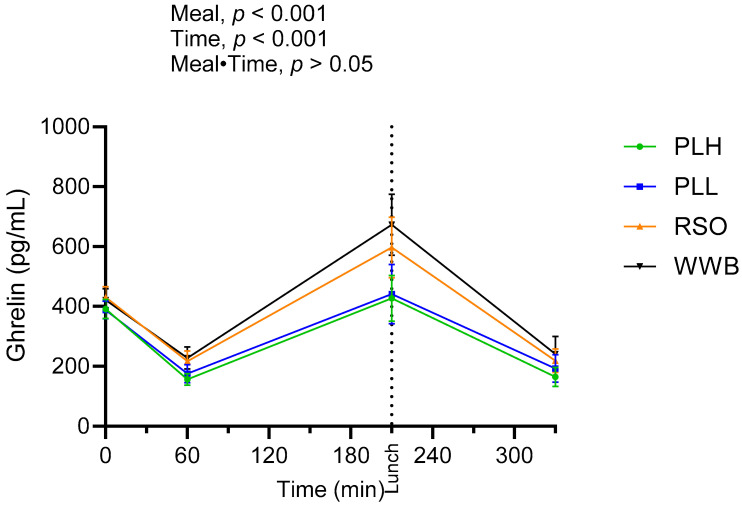
Mean concentration of ghrelin after breakfast and standardized lunch meal. Values are expressed as means ± SEM, with a sample size of n = 20. Repeated measures; mixed model in SAS. WWB, white wheat bread; PLH, 15.0 g oat polar lipids; PLL, 7.5 g oat polar lipids and 8.3 g rapeseed oil; and RSO, 16.6 g rapeseed oil.

**Figure 6 nutrients-15-04389-f006:**
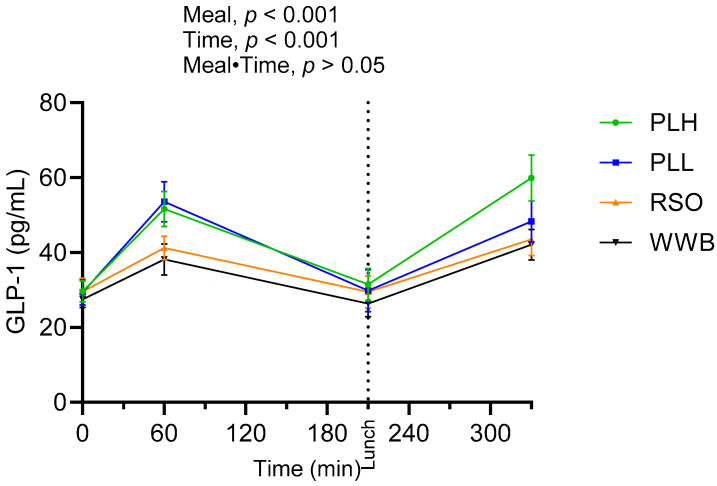
Mean concentration of GLP-1 after breakfast followed by standardized lunch meal. Values are expressed as means ± SEM, with a sample size of n = 20. Repeated measures; mixed model in SAS. WWB, white wheat bread; PLH, 15.0 g oat polar lipids; PLL, 7.5 g oat polar lipids and 8.3 g rapeseed oil; and RSO, 16.6 g rapeseed oil.

**Figure 7 nutrients-15-04389-f007:**
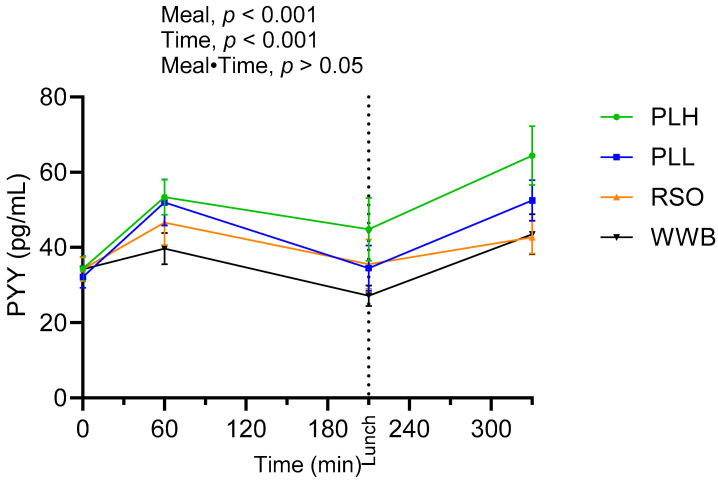
Mean concentration of PYY after breakfast and standardized lunch meal. Values are expressed as means ± SEM, with a sample size of n = 20. Repeated measures; mixed model in SAS. WWB, white wheat bread; PLH, 15.0 g oat polar lipids; PLL, 7.5 g oat polar lipids and 8.3 g rapeseed oil; and RSO, 16.6 g rapeseed oil.

**Figure 8 nutrients-15-04389-f008:**
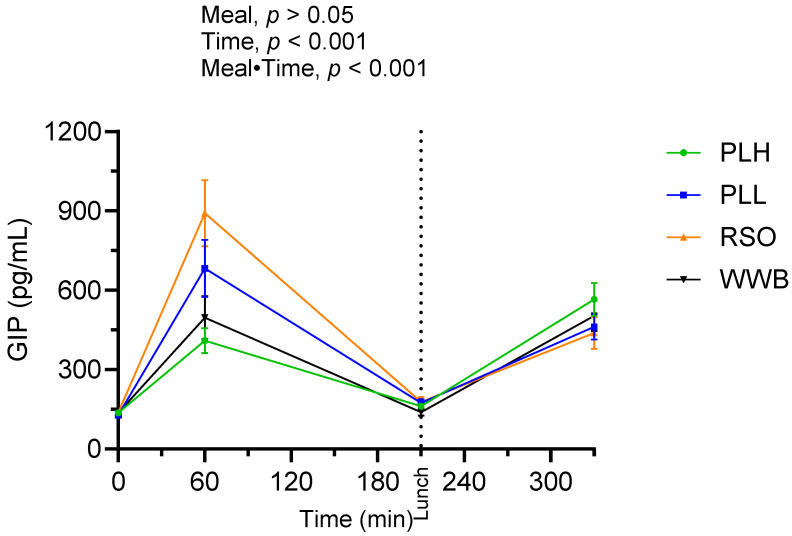
Mean concentration of GIP after breakfast followed by standardized lunch meal. Values are expressed as means ± SEM, with a sample size of n = 20. Repeated measures; mixed model in SAS. WWB, white wheat bread; PLH, 15.0 g oat polar lipids; PLL, 7.5 g oat polar lipids and 8.3 g rapeseed oil; and RSO, 16.6 g rapeseed oil.

**Table 1 nutrients-15-04389-t001:** Formulation of the test and reference meals.

Meals		Meal Compositions
Reference Meal ^1^	WWB	120 g white wheat bread, no added oil, 260 mL water.
Test Meals ^1^	PLH	120 g WWB, 16.6 g of 90% enriched oat polar lipid, 260 ^2^ mL water.
	PLL	120 g WWB, 8.3 g of 90% enriched oat polar lipid, 8.3 g rapeseed oil and 260 ^2^ mL water.
	RSO	120 g WWB, 16.6 g rapeseed oil, 260 mL water.
Standardized Lunch	120 g white wheat bread, 100 g meatballs, 250 mL water	

^1^ WWB, white wheat bread containing 50 g available carbohydrates; PLH, 15.0 g oat polar lipids; PLL, 7.5 g oat polar lipids and 8.3 g rapeseed oil; RSO, 16.6 g rapeseed oil. ^2^ Including 10 mL water used for spread preparation.

**Table 2 nutrients-15-04389-t002:** Macronutrient composition of breakfast and lunch meals ^1^.

	Breakfast	Standardized Lunch
Carbohydrates (g)	50	58
Fat (g)	16.6	18.5
Protein (g)	8.5	21.5
Calories (kcal)	385	485

^1^ In accordance with the manufacturer’s reported nutritional information.

**Table 3 nutrients-15-04389-t003:** Blood glucose and insulin concentrations at fasting, after consumption of test products at breakfast (0–120 min) and a standardized lunch meal (210–330 min) ^1^.

Test Variables	WWB	RSO		PLL		PLH	
			%∆ ^2^		%∆ ^2^		%∆ ^2^
**Glucose**							
**Fasting blood glucose (mmol/L)**	4.91 ± 0.08 ^a^	4.92 ± 0.06 ^a^	0.20	4.92 ± 0.07 ^a^	0.30	4.97 ± 0.06 ^a^	1.32
**Blood glucose at 210, i.e., directly prior to the std. lunch (mmol/L)**	4.67 ± 0.10 ^a^	4.78 ± 0.09 ^a^	2.35	4.85 ± 0.12 ^a^	3.96	4.77 ± 0.09 ^a^	2.24
**Blood glucose iAUC 0–120 min (mmol·min/L)**	165.60 ± 14.90 ^a^	133.70 ± 12.70 ^ab^	−19.26	102 ± 10.40 ^bc^	−38.40	83.8 ± 11.10 ^c^	−49.39
**Blood glucose iAUC 210–330 min (mmol·min/L)**	133.20 ± 10.80 ^a^	118.76 ± 8.77 ^a^	−10.84	113.9 ± 10.2 ^a^	−14.48	78.90 ± 10.20 ^b^	−40.76
**Blood glucose iAUC 0–330 min (mmol·min/L)**	322.40 ± 21.30 ^a^	278.50 ± 19.50 ^ab^	−13.61	239.50 ± 18.50 ^b^	−25.71	171.70 ± 17.10 ^c^	−46.74
**Insulin**							
**Fasting blood insulin (nmol/L)**	0.04 ± 0.007 ^a^	0.04 ± 0.004 ^a^	−5.42	0.05 ± 0.006 ^a^	8.70	0.05 ± 0.006 ^a^	7.06
**Blood insulin at 210 min, i.e., directly prior to std. lunch (nmol/L)**	0.04 ± 0.007 ^a^	0.05 ± 0.009 ^a^	15.6	0.04 ± 0.005 ^a^	0	0.06 ± 0.009 ^a^	21.70
**Insulin iAUC 0–120 (nmol·min/L)**	18.05 ± 2.31 ^a^	17.54 ± 2.19 ^a^	−2.82	14.31 ± 1.71 ^ab^	−20.72	11.19 ± 1.46 ^b^	−38.00
**Insulin iAUC 210–330 (nmol·min/L)**	18.66 ± 2.14 ^ab^	20.13 ± 2.49 ^a^	7.87	18.51 ± 2.13 ^ab^	−0.80	16.68 ± 1.82 ^b^	−10.61
**Insulin iAUC 0–330 (nmol·min/L)**	40.73 ± 4.94 ^a^	41.98 ± 5.13 ^a^	3.06	35.69 ± 3.81 ^ab^	−12.37	30.55 ± 3.37 ^b^	−24.99

^1^ Data are reported as means ± SEM, n = 20. Different superscript letters indicate statistically significant differences between values in the same row, *p* < 0.05 (ANOVA, followed by Tukey’s test). ^2^ The percentage change is obtained as the difference from the WWB. WWB, white wheat bread; PLH, 15.0 g oat polar lipids; PLL, 7.5 g oat polar lipids and 8.3 g rapeseed oil; and RSO, 16.6 g rapeseed oil. iAUC—incremental area under curve.

**Table 4 nutrients-15-04389-t004:** TG and FFA after breakfast (0–210 min) and a standardized lunch (210–330 min) ^1^.

Test Variables	WWB	RSO		PLL		PLH	
Triglycerides (TG)			%∆ ^2^		%∆ ^2^		%∆ ^2^
Fasting TG (mmol/L)	0.65 ± 0.05 ^a^	0.71 ± 0.05 ^a^	8.53	0.68 ± 0.06 ^a^	3.68	0.62 ± 0.04 ^a^	−5.57
TG AUC 0–210 min (mmol·min/L)	145.40 ± 11 ^a^	188.70 ± 15.40 ^b^	29.77	173.40 ± 16.4 ^ab^	19.25	155.70 ± 11 ^a^	7.08
TG AUC 210–330 min (mmol·min/L)	95.92 ± 6.70 ^a^	123.14 ± 8.97 ^b^	28.37	117.5 ± 10.10 ^bc^	22.49	102.69 ± 6.95 ^ac^	7.05
Free Fatty Acids							
Fasting FFA (mmol/L)	0.53 ± 0.03 ^a^	0.45 ± 0.03 ^a^	−15.40	0.46 ± 0.03 ^a^	−14.31	0.46 ± 0.02 ^a^	−14.38
FFA AUC 0–210 min (mmol·min/L)	59.36 ± 3.42 ^a^	62.52 ± 3.32 ^a^	5.32	58.01 ± 3.10 ^a^	−2.27	60.18 ± 3.34 ^a^	1.38
FFA AUC 0–210 min (mmol·min/L)	31.46 ± 1.96 ^a^	34.29 ± 2.50 ^a^	8.99	31.57 ± 1.87 ^a^	0.34	31.96 ± 1.83 ^a^	1.58

^1^ Data are reported as means ± SEM, with a sample size of n = 20. Different superscript letters indicate statistically significant differences between values in the same row, *p* < 0.05 (ANOVA, followed by Tukey’s test). ^2^ The percentage change is obtained as the difference from the WWB. WWB, white wheat bread; PLH, 15.0 g oat polar lipids; PLL, 7.5 g oat polar lipids and 8.3 g rapeseed oil; and RSO, 16.6 g rapeseed oil. iAUC—incremental area under curve.

**Table 5 nutrients-15-04389-t005:** Concentrations of plasma ghrelin, GLP-1, PYY, and GIP after breakfast followed by standardized lunch ^1^.

Test Variables	WWB	RSO		PLL		PLH	
			%∆ ^2^		%∆ ^2^		%∆ ^2^
Fasting plasma ghrelin (pg/mL)	421.3 ± 38.50 ^a^	430.80 ± 36.50 ^a^	2.25	388 ± 29.10 ^a^	−7.9	390.90 ± 32.10 ^a^	−7.21
Ghrelin AUC 0–210 (pg·min/mL)	87,014 ± 11,233 ^a^	80,501 ± 10,469 ^a^	−7.48	61,068 ± 9501 ^b^	−29.81	60,148 ± 7948 ^b^	−30.87
Ghrelin at 330 min (pg/mL)	236.30 ± 52.90 ^a^	213.40 ± 39.3 ^a^	−9.69	207.60 ± 45.80 ^a^	−12.14	164.70 ± 31.90 ^a^	−30.3
Fasting plasma GLP-1 (pg/mL)	27.52 ± 2.16 ^a^	26.06 ± 2.49 ^a^	−5.3	29.34 ± 3.37 ^a^	6.64	29.63 ± 2.84 ^a^	7.67
GLP-1 AUC 0–210 (pg·min/mL)	6378 ± 599 ^a^	6875 ± 554 ^a^	7.79	8743 ± 862 ^b^	37.08	9296 ± 583 ^b^	45.75
GLP-1 at 330 min (pg/mL)	42.12 ± 4.10 ^a^	42.3 ± 4.27 ^a^	0.42	48.29 ± 5.49 ^ab^	14.64	60.10 ± 6.09 ^b^	42.68
Fasting plasma PYY (pg/mL)	32.01 ± 3.09 ^a^	34.26 ± 3.19 ^a^	7.02	32.2 ± 2.93 ^a^	0.59	36.61 ± 3.36 ^a^	14.37
PYY AUC 0–210 (pg·min/mL)	6747 ± 581 ^a^	8595 ± 1030 ^b^	27.38	9011 ± 1064 ^bc^	33.55	10,481 ± 1036 ^c^	55.34
PYY at 330 min (pg/mL)	41.64 ± 5.18 ^a^	42.67 ± 4.38 ^a^	2.47	52.5 ± 5.43 ^a^	26.08	66.32 ± 7.67 ^b^	59.26
Fasting plasma GIP (pg/mL)	136.93 ± 8.18 ^a^	137.35 ± 9.65 ^a^	0.3	127.31 ± 5.94 ^a^	−7.02	136.4 ± 11.20 ^a^	−0.38
GIP AUC 0–210 (pg·min/mL)	66,561 ± 8999 ^ac^	111,032 ± 13,836 ^b^	66.81	88,481 ± 11,853 ^ab^	32.93	59,218 ± 5365 ^c^	−11.03
GIP at 330 min (pg/mL)	503.7 ± 58.60 ^a^	438.6 ± 60.1 ^ab^	−12.92	461.10 ± 47.90 ^ab^	−8.45	565.9 ± 62.30 ^b^	12.34

^1^ Data are reported as means ± SEM, with a sample size of n = 20. Different superscript letters indicate statistically significant differences between values in the same row, *p* < 0.05 (ANOVA, followed by Tukey’s test). ^2^ The percentage change is obtained as the difference from the WWB. WWB, white wheat bread; PLH, 15.0 g oat polar lipids; PLL, 7.5 g oat polar lipids and 8.3 g rapeseed oil; and RSO, 16.6 g rapeseed oil. AUC—area under curve.

## Data Availability

The datasets analyzed during this study are available from the corresponding author on reasonable request.

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
