# Peer review of "Inclusion of Oat Polar Lipids in a Solid Breakfast Improves Glucose Tolerance, Triglyceridemia, and Gut Hormone Responses Postprandially and after a Standardized Second Meal: A Randomized Crossover Study in Healthy Subjects"

_nutrients, 2023, doi:10.3390/nu15204389_

Round 1

Reviewer 1 Report

The article “Inclusion of Oat Polar Lipids in a Solid Breakfast Improves Glucose Tolerance, Triglyceridemia and Gut Hormone Responses Postprandially and after a Standardized Second meal: A Randomized Crossover Study in Healthy Subjects” has focused on interesting problem – the metabolic effects of different doses of oat polar lipids incorporated in a solid breakfast meal. The main results from the research showed the beneficial effects of polar lipids from oats on acute and second meal postprandial glycaemic control, blood lipids, and gastrointestinal hormones in healthy volunteers. Nevertheless, the Reviewer has some objections:

1. Authors presented the results obtained in small sample size (only 20 participants), thus the study could be treated rather as a pilot. In future studies it would be also worth to extend it to other populations. The simple size calculations should be given in Methods section.

2. The main objection occurred the errors in literature citation in the text – from the Material and Methods section till the Discussion section. Instead the literature citation or number there is “Error! Reference source not found”. The references should be corrected in the whole text.

3. The exclusion and inclusion criteria should be given.

4. There is lack of the citation of Tables and Figures in the text.

5. How much of blood was collected? It should be specified.

6. Why the blood was frozen in -40°C instead -80°C?

7. Line 189: The method description should be more precise.

8. Line 222: Furthermore, the breakfast postprandial glucose concentrations (iAUC 0-120 min) were reduced following PLL compared to WWB – please explain.

9. The Discussion section could be expanded. More information concerning the mechanism linking the polar lipid action and metabolic health could be given. For example how they interact with the gastrointestinal hormones? Are there any mechanisms?

10. There is a question about the practical dimension of the study – please discuss it. How many polar lipids should be take to improve metabolic health?

11. There is lack of the strengths of the study.

12. It is recommended to review the entire manuscript and references for spelling and editorial errors.

13. English and scientific language should be improved.

English and scientific language should be improved. The manuscript should be check by a native speaker. 

Author Response

MDPI

Nutrients Editorial Office

Dear Reviewer,

Thank you for reviewing our manuscript titled “Inclusion of Oat Polar Lipids in a Solid Breakfast Improves Glucose Tolerance, Triglyceridemia and Gut Hormone Responses Postprandially and after a Standardized Second meal: A Randomized Crossover Study in Healthy Subjects” to Nutrients (Manuscript ID: nutrients-2662084). We appreciate the time and effort that you have dedicated to providing your valuable feedback on our manuscript. We have been able to incorporate changes to reflect most of the suggestions provided by you. We have highlighted the changes within the manuscript.

Here is a point-by-point response to the reviewers’ comments and concerns.

Comments from Reviewer

Overall Comments: Comments and Suggestions for Authors

The article “Inclusion of Oat Polar Lipids in a Solid Breakfast Improves Glucose Tolerance, Triglyceridemia and Gut Hormone Responses Postprandially and after a Standardized Second meal: A Randomized Crossover Study in Healthy Subjects” has focused on interesting problem – the metabolic effects of different doses of oat polar lipids incorporated in a solid breakfast meal. The main results from the research showed the beneficial effects of polar lipids from oats on acute and second meal postprandial glycaemic control, blood lipids, and gastrointestinal hormones in healthy volunteers. Nevertheless, the Reviewer has some objections.

Response: We appreciate the time and effort that you have dedicated to providing your valuable feedback on our manuscript.

Specific comments:

Comment 1: Authors presented the results obtained in small sample size (only 20 participants), thus the study could be treated rather as a pilot. In future studies it would be also worth to extend it to other populations. The simple size calculations should be given in Methods section.

Response: You have raised an important point here. The primary outcome measure of the study was incremental changes of blood glucose concentration. We have added a new sub-section in the materials and methods called “Power Calculation”. On that basis the 20 test subjects included were enough for detecting a minimum of 15% change in blood glucose response (measured as iAUC) (p5, line 219-223).

Comment 2: The main objection occurred the errors in literature citation in the text – from the Material and Methods section till the Discussion section. Instead the literature citation or number there is “Error! Reference source not found”. The references should be corrected in the whole text.

Response: Thank you for pointing this out. The original submitted version did not contain “Error! Reference source not found”. We believe that the editor’s office made some format change and, due to that, the Microsoft word cross-reference source was broken. We have revised the issue and fixed it.

Comment 3: The exclusion and inclusion criteria should be given.

Response: Thank you for the comment. The inclusion and exclusion criteria are now clearly described. (p2 line 92-94).

Comment 4: There is lack of the citation of Tables and Figures in the text.

Response: Thank you for the suggestion. Citation of Tables and Figures were fixed and revised the whole manuscript.

Comment 5: How much of blood was collected? It should be specified.

Response: We appreciate your suggestion and we have specified the amount of blood collected (P4, line 152-155).

Comment 6: Why the blood was frozen in -40°C instead -80°C?

Response: Previous studies from this research group have shown that freezing serum and plasma samples at -40°C is sufficient to preserve the integrity of the analytes investigated.

Comment 7: Line 189: The method description should be more precise.

Response: Requested description was added (p5 line 192- 197).

Comment 8: Line 222: Furthermore, the breakfast postprandial glucose concentrations (iAUC 0-120 min) were reduced following PLL compared to WWB – please explain

Response: We have changed the text to make the message clearer (p5, line 234-236).

Comment 9: The Discussion section could be expanded. More information concerning the mechanism linking the polar lipid action and metabolic health could be given. For example how they interact with the gastrointestinal hormones? Are there any mechanisms?

Response: Thank you for this suggestion and we have incorporated more text (p13-14, line 420-429).

Comment 10: There is a question about the practical dimension of the study – please discuss it. How many polar lipids should be take to improve metabolic health?

Response: Revised as requested (p14, line 453-457).

Comment 11: There is lack of the strengths of the study.

Response: The whole paragraph (p14, line 460-6-465) was re-written in order to emphasize the strengths and weaknesses of the study.

Comment 12: It is recommended to review the entire manuscript and references for spelling and editorial errors.

Response: Thank you for this suggestion and we agree on this. We have revised the whole manuscript.

Comment 13: English and scientific language should be improved.

Response: Revised as requested.

Sincerely,

Mohammad Mukul Hossain

Lund University, Sweden.

Reviewer 2 Report

The manuscript named with “Inclusion of Oat Polar Lipids in a Solid Breakfast Improves Glucose Tolerance, Triglyceridemia and Gut Hormone Responses Postprandially and after a Standardized Second meal: A Randomized Crossover Study in Healthy Subjects” is to evaluate effects of oat polar lipids in a solid food matrix on acute and second meal glucose tolerance, blood lipids, and concentrations of gut derived hormones. The results showed some benefit effect on glycemia responses and some regulating effect of GI hormones and metablic proteins. It is interesting topic for nutritional research field.  But there are some things that need to be clarified and improved. 

Please describe more about the different points in the background, at which Oat Polar Lipids affect glycolipid metabolism in the context of solid and liquid foods, including the differences between liquid foods and solid foods + water drinking, so this study is to be carried out. Please give the study assumptions. 

In the Results, please give the more detailed descriptions in Figures and more detailed descriptions and data in Tables. 

Please give some interpretation on TG AUC increased in the PLH group compared with WWB. 

Minors: 

Line 103: It is unclear enough that the quatitative of PL were put in each group. Please check the sentence “added lipids consisted of WWB with either 1: 15.0 g PL (PL higher (H) amounts), 2: 7.5 g PL”? 

Line 104: Since there were two rapeseed oil group with rapeseed oil in ratio 50/50 (PL lower (L) amounts), or 3: 16.6 g, there are five groups (four test and one control groups) in this study?

Line 107: please directly give the capacity of the water bottle to hold water or give the amount of water to drink. 

Line 112: Please give reference to “lunch meal contained a total calorie value of 485 kcal”. 

Line 134: How long to refrain from.........,for all of these conditions? 

How many milliliters of blood are drawn each time? 

Line 214-216: Please clarify the sentence Data can be found in Error! Reference source not found. (glucose and insulin), Error! Reference source not found. (TG and FFA) and Error! Reference source not found. (Ghrelin, GLP-1, PYY and GIP) and others. 

What the meaning of ab in every tables? Please give the notes.  

The English Language is reasonable.  

Author Response

MDPI

Nutrients Editorial Office

Dear Reviewer,

Thank you for reviewing our manuscript titled “Inclusion of Oat Polar Lipids in a Solid Breakfast Improves Glucose Tolerance, Triglyceridemia and Gut Hormone Responses Postprandially and after a Standardized Second meal: A Randomized Crossover Study in Healthy Subjects” to Nutrients (Manuscript ID: nutrients-2662084). We appreciate the time and effort that you have dedicated to providing your valuable feedback on our manuscript. We have been able to incorporate changes to reflect most of the suggestions provided by you. We have highlighted the changes within the manuscript.

Here is a point-by-point response to the reviewers’ comments and concerns.

Comments from Reviewer

Overall Comment: The manuscript named with “Inclusion of Oat Polar Lipids in a Solid Breakfast Improves Glucose Tolerance, Triglyceridemia and Gut Hormone Responses Postprandially and after a Standardized Second meal: A Randomized Crossover Study in Healthy Subjects” is to evaluate effects of oat polar lipids in a solid food matrix on acute and second meal glucose tolerance, blood lipids, and concentrations of gut derived hormones. The results showed some benefit effect on glycemia responses and some regulating effect of GI hormones and metablic proteins. It is interesting topic for nutritional research field.  But there are some things that need to be clarified and improved.

Response: We would like to thank reviewer for such nice comments. We appreciate the time and effort that you have dedicated to providing your valuable feedback on our manuscript.

Specific comments:

Comment 1: Please describe more about the different points in the background, at which Oat Polar Lipids affect glycolipid metabolism in the context of solid and liquid foods, including the differences between liquid foods and solid foods + water drinking, so this study is to be carried out. Please give the study assumptions.

Response: It is clear that the physical form of a meal has an important impact on the digestion and absorption of macronutrients. Food matrix has particular importance in the case of fat. Therefore, the current investigation using a solid meal is of relevance. Combining a solid meal with drinking water does not have the same impact on digestibility as the intake of already dissolved/dispersed macronutrients in a liquid meal. We have revised the introduction as suggested. (p2, line 72-78).

Comment 2: In the Results, please give the more detailed descriptions in Figures and more detailed descriptions and data in Tables.

Response: We are sorry for the confusion. We don’t understand the comment since all figures and tables contained corresponding captions and footnotes. No action was taken on this regard.

Comment 3: Please give some interpretation on TG AUC increased in the PLH group compared with WWB.

Response: The polar lipid preparation used in this study contained 10% TG which explains the statistically non-significant blood TG increase after consumption. That was indicated in the results section (p3, line 103).

Minor Comments:

Comment 4: Line 103: It is unclear enough that the quatitative of PL were put in each group. Please check the sentence “added lipids consisted of WWB with either 1: 15.0 g PL (PL higher (H) amounts), 2: 7.5 g PL”?

Response: We have changed the text in order to provide a clearer description of the test meals (p3, line 110-112).

Comment 5: Line 104: Since there were two rapeseed oil group with rapeseed oil in ratio 50/50 (PL lower (L) amounts), or 3: 16.6 g, there are five groups (four test and one control groups) in this study?

Response: There are four groups in the study (three test and one control). The rapeseed oil was added to PLL to get equivalent amount of total fat. The description was modified (p3, line 110).

Comment 6: Line 107: please directly give the capacity of the water bottle to hold water or give the amount of water to drink

Response: Thank you for the suggestion. We have added the volume of water (p3 line 114).

Comment 7: Line 112: Please give reference to “lunch meal contained a total calorie value of 485 kcal”.

Response: The reference is now given as suggested.

Comment 8: Line 134: How long to refrain from.........,for all of these conditions?

Response: The time was mentioned in page 3 line 132.

Comment 9: How many milliliters of blood are drawn each time?

Response: Revised as suggested (p4, line 151-154).

Comment 10: Line 214-216: Please clarify the sentence “Data can be found in Error! Reference source not found. (glucose and insulin), Error! Reference source not found. (TG and FFA) and Error! Reference source not found. (Ghrelin, GLP-1, PYY and GIP) and others.

Response: Thank you for pointing this out. The original submitted version did not contain “Error! Reference source not found”. We believe that the editor’s office made some format change and, due to that, the Microsoft word cross-reference source was broken. We have revised the issue and fixed it.

Comment 11: What the meaning of “ab” in every tables? Please give the notes

Response: The meaning of ‘ab’ superscript was defined in the original manuscript and are found in the footnotes as “Values in the same row with different superscript letters are significantly different, P < 0.05”. (p.7, line 262).

Sincerely,

Mohammad Mukul Hossain

Lund University, Sweden.
